# Urine Metabolomic Profiling and Machine Learning in Autism Spectrum Disorder Diagnosis: Toward Precision Treatment

**DOI:** 10.3390/metabo15050332

**Published:** 2025-05-16

**Authors:** Shula Shazman, Julie Carmel, Maxim Itkin, Sergey Malitsky, Monia Shalan, Eyal Soreq, Evan Elliott, Maya Lebow, Yael Kuperman

**Affiliations:** 1Department of Mathematics and Computer Science, The Open University of Israel, Raanana 4353701, Israel; 2Azrieli Faculty of Medicine, Bar Ilan University, Safed 1311502, Israel; jc.book@gmail.com (J.C.); evanmelliott@gmail.com (E.E.); 3Metabolic Profiling Unit, Life Sciences Core Facilities, Weizmann Institute of Science, Rehovot 7610001, Israel; maxim.itkin@weizmann.ac.il (M.I.); sergey.malitsky@weizmann.ac.il (S.M.); 4Ziv Medical Center, Safed 1311502, Israel; monia.s@ziv.gov.il; 5Department of Brain Science, Faculty of Medicine, Imperial College London, London SW3 6LY, UK; eyalsoreq@gmail.com; 6Care, Research & Technology Centre, UK Dementia Research Institute, London NW1 3BT, UK; 7The NIHR Imperial, Biomedical Research Centre, London W12 0NN, UK; 8ANeustart, Ltd., Rishon LeZion 7526088, Israel

**Keywords:** autism spectrum disorder, metabolomics, urine metabolites, machine learning, diagnostic biomarkers, random forest, endogenous metabolites, exposome, liquid chromatography-mass spectrometry, machine learning classifiers

## Abstract

Background: Autism spectrum disorder (ASD) diagnosis traditionally relies on behavioral assessments, which can be subjective and often lead to delayed identification. Recent advances in metabolomics and machine learning offer promising alternatives for more objective and precise diagnostic approaches. Methods: First-morning urine samples were collected from 52 children (32 with ASD and 20 neurotypical controls), aged 5.04 ± 1.87 and 5.50 ± 1.74 years, respectively. Using liquid chromatography-mass spectrometry (LC-MS), 293 metabolites were identified and categorized into 189 endogenous and 104 exogenous metabolites. Various machine learning classifiers (random forest, logistic regression, random tree, and naïve Bayes) were applied to differentiate ASD and control groups through 10-fold cross-validation. Results: The random forest classifier achieved 85% accuracy and an area under the curve (AUC) of 0.9 using all 293 metabolites. Classification based solely on endogenous metabolites yielded 85% accuracy and an AUC of 0.86, whereas using exogenous metabolites alone resulted in lower performance (71% accuracy and an AUC of 0.72). Conclusion: This study demonstrates the potential of urine metabolomic profiling, particularly endogenous metabolites, as a complementary diagnostic tool for ASD. The high classification accuracy highlights the feasibility of developing assistive diagnostic methods based on metabolite profiles, although further research is needed to link these profiles to specific behavioral characteristics and ASD subtypes.

## 1. Introduction

Autism spectrum disorder (ASD) is a pervasive neurodevelopmental condition characterized by deficits in social interaction, communication, and repetitive behaviors [1]. The increasing prevalence of ASD, which is estimated to affect 1 in 36 children in the United States [2], underscores the urgent need for more accurate diagnostic methods and personalized treatment strategies. The traditional diagnostic approaches, which rely heavily on behavioral assessments, are often subjective and may lead to delayed or inaccurate diagnoses [3].

Metabolomic profiling involves the comprehensive analysis of small molecules in biological samples, including urine [4,5]. Metabolomic studies of the urine of children with autism highlight the heterogeneity in the autism spectrum and have revealed several metabolites that discriminate between children with autism and neurotypical children [6,7,8,9,10,11]. These metabolites have the potential to identify unique metabolic signatures associated with ASD [8]. Discriminant metabolites identified originate from both exogenous and endogenous sources and include derivatives or precursors to amino acids, neurotransmitters, essentials of the Krebs cycle, metabolites of microbial or exposome origins, and other biological pathways [6,7,8,12]. In some cases, urinary metabolite levels are correlated with blood metabolite levels [13]. These metabolic differences are influenced by diet, gut dysbiosis, oxidative stress, tryptophan metabolism, mitochondrial dysfunction, and other underlying biological pathways that contribute to the etiology of autism or to subsets of children with autism [14,15,16,17,18,19].

Recent advancements in systems biology and metabolomics offer promising avenues for improving the accuracy and timeliness of ASD diagnosis. Integrating metabolomic data with machine learning techniques further enhances the potential for precise ASD diagnosis and treatment. Machine learning algorithms can analyze complex metabolic data to identify patterns and correlations that may not be apparent through traditional statistical methods. This approach has shown promise in characterizing and distinguishing ASD subtypes and correlating metabolic profiles with symptom severity [8], paving the way for precision medicine in ASD treatment.

In this context, the current study explores the intersection of metabolomic profiling and machine learning in the diagnosis of ASD. We investigated the urine metabolomic basis of ASD in an Israeli cohort and demonstrated that the urine metabolite profile could classify children with ASD with an accuracy of 85%. Various machine learning classifiers were utilized, including the random forest, logistic regression, J48 pruned tree, and naïve Bayes classifiers. The random forest is a robust and flexible method applicable to a wide range of tasks, whereas a simpler model, such as the J48 pruned tree, is faster and easier to interpret. Logistic regression yields interpretable results and performs well with multidimensional data, whereas naïve Bayes is efficient but relies on a strong independence assumption. These findings highlight the potential of these technologies to improve diagnostic accuracy and, ultimately, contribute to personalized therapeutic strategies. By harnessing the power of metabolomics and advanced data analytics, we are advancing toward a future where ASD diagnosis and treatment are tailored to everyone’s unique metabolic and phenotypic profiles.

## 2. Methods

### 2.1. Participants

Children diagnosed with ASD: children who were diagnosed with autism as per the Diagnostic and Statistical Manual of Mental Disorders 5 (DSM 5) criteria. Control children: children with no ASD diagnosis. In both groups, children with a known genetic disorder (Fragile X syndrome, Rett syndrome, or metabolic or other monogenic disorders) or children chronically treated daily or weekly with any medications for the past month (such as psychiatric or steroids or medication for chronic illnesses) except for methylphenidate or asthma-related inhalant steroids were excluded. In the case of antibiotic usage, urine collection was postponed until three months after treatment.

Informed consent was obtained from all the participants’ parents. This study was approved by the Ziv Medical Center in Tzfat, Israel.

Overall, 52 children were recruited: 20 controls (2 females), aged 5.04 + 1.87 years, and 32 ASD patients (5 females), aged 5.50 + 1.74 years.

### 2.2. Urine Collection

First-morning urine was collected into a urine cup or a “nurse’s hat” and immediately divided into 1 mL aliquots in Eppendorf tubes. The urine was frozen at −18 °C and stored at −80 °C until analysis.

### 2.3. Urine Analysis

The extraction and analysis of polar metabolites were performed as previously described by Gnainsky et al. (2022) [20], with slight modifications:

The urine samples (100 µL) were extracted with 1 mL of a precooled (−20 °C) homogenous methanol–methyl-tert-butyl-ether (MTBE) 1:3 (*v*/*v*) mixture. The tubes were vortexed and then sonicated for 30 min in an ice-cold sonication bath (taken for a brief vortex every 10 min). A UPLC-grade water (DDW)–methanol (3:1, *v*/*v*) solution (0.5 mL), containing C13- and N15-labeled amino acid standard mixtures (Sigma-Aldrich, Burlington, MA, USA, 767964) (1:1500), was added to the tubes, which were then vortexed and centrifuged. The upper organic phase was discarded. The polar phase was re-extracted with 0.5 mL of MTBE. The lower polar phase, which was used for polar metabolite analysis, was dried in a SpeedVac for one hour, lyophilized, and then stored at −80 °C. Dry polar samples were resuspended in 120 µL of methanol–DDW (50:50) and centrifuged (20,800 rcf) twice, and 70 µL was transferred to the HPLC vials for injection.

### 2.4. LC-MS Analysis of Polar Metabolites

Metabolic profiling of the polar phase was performed using an Acquity I-Class UPLC system coupled to a Q Exactive Plus Orbitrap™ mass spectrometer (Thermo Fisher Scientific, Waltham, MA, USA). Chromatographic separation was achieved using a SeQuant ZIC-pHILIC column (150 mm × 2.1 mm) with a SeQuant guard column (20 mm × 2.1 mm) (Merck Group, Darmstadt, Germany). The mobile phases consisted of solvent B (acetonitrile) and solvent A (20 mM ammonium carbonate with 0.1% ammonium hydroxide in DDW–acetonitrile (80:20, *v*/*v*)). The flow rate was set to 200 µL/min, and the column temperature was maintained at 45 °C.

The LC gradient was programmed as follows: 0–2 min, 75% B; 2–14 min, linear decrease to 25% B; 14–18 min, hold at 25% B; 18–19 min, linear increase to 75% B; 19–23 min, hold at 75% B. The injection volume was 2 µL.

Data were acquired in the *m*/*z* range of 70–1050 using heated electrospray ionization (HESI) in negative mode. The ion source parameters were as follows: capillary temperature, 325 °C; spray voltage, 3.25 kV; sheath gas flow rate, 40; auxiliary gas flow rate, 10 (arbitrary units); and auxiliary gas temperature, 50 °C. MS1 spectra were collected at a resolution of 35,000 FWHM. Data-dependent MS/MS acquisition was performed with an isolation window of 3 *m*/*z* and a resolution of 17,500 FWHM.

The instrument was calibrated every 48 h during operation to ensure mass accuracy. Internal standards were added to all samples to normalize for potential variability in sample preparation and instrument performance. Additionally, throughout the analysis, a mixture of external standards, a pooled quality control sample, and a “super sample” were regularly injected to monitor signal stability and correct for intensity drift. These controls also served to validate metabolite identification by comparison with an in-house library of retention times and MS/MS spectra.

### 2.5. Creatinine Normalization

Creatinine levels in urine samples were measured using the same polar LC-MS method. A calibration curve was prepared using 13 creatinine concentrations ranging from 0.2 to 1000 µg/mL to confirm linearity within the physiological range. Metabolite intensities were normalized to creatinine to account for variability in urine concentration.

### 2.6. Data Processing and Identification

Raw LC-MS data were processed using Compound Discoverer software (v3.3, Thermo Fisher Scientific). Detected compounds were putatively identified based on accurate mass (less than 5 ppm), retention time, isotope pattern, and MS/MS fragmentation, and were verified against an in-house spectral library. Relative levels of detected metabolites were normalized to both internal standards and urinary creatinine.

### 2.7. Datasets

A total of 293 polar urine metabolites were collected from first-morning urine samples of 32 children diagnosed with ASD and 20 age-matched neurotypical controls.

The 293 polar urine metabolites were divided into two groups based on their potential source, and three datasets were run:(1)Endogenous metabolites: A total of 189 metabolites in the body can be endogenously produced or obtained from endogenous metabolism, such as amino acids or participants in the Krebs cycle.(2)Exogenous metabolites: 104 metabolites originating from the exposome, meaning metabolites that exclusively originate from the environment and are subsequently processed by the body. These include food metabolites, metabolites produced by the microbiome, and metabolites present in the surroundings, such as food packaging, cosmetics, and pharmaceuticals.(3)All 293 metabolites were combined.

### 2.8. Machine Learning Classifiers

Different machine learning classifiers, including random forest, logistic regression, J48 pruned tree, and naïve Bayes, were used, utilizing WEKA 3.8.6 software [21]. All classification models were implemented using the WEKA platform 3.8.6, with default (unmodified) hyperparameters applied to all algorithms. We chose to use these base (unmodified) hyperparameters to provide a standardized evaluation across models without introducing potential advantages through model-specific tuning. The testing approach employed was 10-fold cross-validation, where the model was built using nine-tenths of the data and tested on the remaining one-tenth. This process was repeated 10 times.

In parallel, conventional methods like PCA and clustering were used for exploration analysis, highlighting data structure and group separation. Although PCA and clustering are useful for identifying patterns and key features, supervised machine learning classifiers were ultimately selected for final classification due to their superior ability to handle complex, high-dimensional data and achieve higher accuracy and AUC values.

### 2.9. Questionnaires

The parents of the children included in the study completed questionnaires, including the Autism Treatment Evaluation Checklist (ATEC) [22], which contains the following domains: sociability, speech/language/communication, health/physical/behavior, and sensory/cognitive awareness. The ATEC was developed to evaluate the impact of intervention broadly. A reduction in score in any subdomain or the total score indicates a reduction in symptom severity. The parents also completed the Strengths and Difficulties Questionnaire (SDQ) [23], which has 25 questions that measure behavioral and emotional difficulties and can be used to assess coping mechanisms with challenges to external and internal stimuli.

### 2.10. Feature Selection

A feature selection analysis was performed using all four classification algorithms. The decision tree based on the J48 pruned tree identified four key features (metabolites). Accordingly, we also retrieved the top four most important features for each of the other classifiers. The results revealed that each classifier selected a different set of four metabolites. This outcome is expected, given the presence of 293 metabolites and the different mechanisms underlying each classifier.

Next, we evaluated the classification performance by using, in turn, the top four metabolites identified by each classifier as input features across all classifiers. The results showed that the best performance across all four classifiers was achieved when using the four metabolites selected by the J48 pruned tree: Glycerate, 3,4-Dihydroxyhydrocinnamic acid, 3,4-Dihydroxybenzoate, and 5-Hydroxyindoleacetate.

Additionally, we tested all 16 top-ranked features (the four selected by each classifier combined), but none achieved classification results as strong as those obtained with the four metabolites selected by the J48 pruned tree. Therefore, we proceeded with the four J48-selected metabolites for subsequent analyses.

## 3. Results and Discussion

### 3.1. Classification Based on All 293 Metabolites

Based on the analysis of the 293 urine polar metabolites, we trained and tested several machine learning classifiers to differentiate between children with ASD and controls (for detailed results, please refer to Appendix A). Among the tested classifiers, random forest achieved the highest AUC (0.90), making it the best at distinguishing between classes, though the J48 pruned tree slightly outperformed it in accuracy (87% vs. 85%). Meanwhile, naïve Bayes and logistic regression performed significantly worse, with lower AUC values (0.61) and lower accuracy rates (52% and 62%). These weaker results stem from naïve Bayes’ unrealistic independence assumption and the linear nature of logistic regression, which struggles with complex, non-linear relationships. In contrast, tree-based models like random forest and J48 excel at handling interactions between features, non-linear relationships, and imbalanced data.

The model built using the J48 pruned decision tree can be visualized as a decision tree consisting of the most important features for distinguishing ASD patients from controls (Figure 1A). As shown in the tree, only two of the 52 children were incorrectly classified.

The most important metabolite for differentiating ASD patients and controls is glycerate (Figure 1B). High urine levels of glycerate can result from either a deficiency of glycerate kinase, which can cause neurological impairment [24], or microbial yeasts such as *Aspergillus* and *Candida* [25]. Additional distinguishing features are 3-4 dihydroxyhydrocinnamic acid, 5-hydroxyindoleacetate, and 4-hydroxy-3-methylbenzoic acid (Figure 1C–E). 3-4 Dihydroxyhydrocinnamic acid (Figure 1C) is a metabolite found after consuming antioxidant-rich foods. 5-Hydroxyindoleacetate (5-HIAA) is a metabolite of serotonin that is metabolized by the enzyme Monoamine Oxidase (MAO). A subset of children with ASD has already been reported to have platelet hyperserotonemia, and high urinary 5-HIAA levels can identify these children [26]. Nevertheless, in our cohort, only a few children had high 5-HIAA values (Figure 1D). 4-Hydroxy-3-methylbenzoic acid is a secondary metabolite possibly resulting from the incomplete breakdown of foods such as poultry (Figure 1E).

While only glycerate levels were significantly different between the ASD and control groups, the variability of the other metabolite levels suggests that there are children with unique sets of values of these metabolites, i.e., extremely high or low, that collectively resulted in better classification when examined as part of the full metabolomic fingerprint.

### 3.2. Classification of 189 Endogenous Metabolites

Next, we trained and tested several machine learning classifiers to differentiate between children with ASD and controls based on the 189 polar urine metabolites that the body may produce endogenously.

Classification based on 189 endogenous metabolites appeared to drive the performance achieved by all metabolites, with the random forest classifier reaching an accuracy of 81% and an AUC of 0.86. Detailed results are presented in Appendix A. For endogenous metabolites, linear models such as logistic regression and naïve Bayes demonstrated improved performance when evaluated on a subset of the dataset, likely due to fewer outliers and simpler class boundaries. Logistic regression’s accuracy increased from 62% to 69%, with its AUC rising from 0.61 to 0.71. Similarly, naïve Bayes showed an AUC improvement from 0.61 to 0.72 but maintained a relatively low accuracy of 54%, likely due to its assumption of feature independence. In contrast, tree-based models like J48 and random forest experienced slight declines in performance on the subset, suggesting that their advantage lies in capturing complex patterns present in the full dataset—patterns that are less prominent in the smaller subset (endogenous metabolites).

The decision tree generated by the J48 classifier for the 189 endogenous metabolites (Figure 2A) is similar to the decision tree for classifying all metabolites (Figure 1A); only two children out of the 52 were incorrectly classified.

As in the decision tree for classifying all metabolites, the first two nodes are glycerate and 3-4 dihydroxyhydrocinnamic. However, the branching continues to present other important endogenous metabolites for distinguishing between ASD patients and controls: Meso-Tartate, N-acetylcysteine, and Pro-Gly.

N-acetylcysteine (NAC) is a derivative of the amino acid cysteine and plays an integral role in glutathione production. Several clinical trials have shown promise in reducing irritability after NAC administration in children with ASD [27,28,29]. The NAC distribution in our cohort was similar between the ASD and control groups (Figure 2B).

Pro-Gly is a dipeptide composed of the amino acids proline and glycine, and it is formed during incomplete protein digestion, suggesting reduced utilization of both components. There are downstream implications, as both proline and glycine are dysregulated in autism. For example, significantly lower levels of proline have been detected in the urine of children with ASD [30], and glycine, which is essential for brain function, acts as an agonist for NMDA receptors and is involved in glutathione production—which is known to be reduced in autism [31]. The Pro-Gly distribution in our cohort was similar between the ASD and control groups (Figure 2C).

Meso-tartate is an isomer produced by the processing of tartaric acid found in foods such as wine and used as a food additive. While tartaric acid levels have been measured in the urine of children with ASD, the results are inconsistent across studies. Some studies reported significantly reduced levels [32], others reported no difference [9], and a case study reported elevated levels, although diet was not evaluated [33]. Though relatively higher levels of meso-tartate were observed in our cohort (*p* = 0.10), the findings were inconclusive and may point to the importance of further independent analysis of this isomer (Figure 2D).

### 3.3. Classification Based on 104 Exogenous Metabolites

Using only the exogenous metabolites resulted in poorer classification than the endogenous metabolites. Detailed results are presented in Appendix A. In the exogenous metabolite’s evaluation, the J48 pruned tree maintained consistent performances across all datasets (AUC of 0.85, accuracy of 83%), highlighting its strong generalization capabilities. Random forest, which previously led in AUC, experienced a moderate drop in both AUC (from 0.90 to 0.72) and accuracy (from 85% to 71%), suggesting that the latest dataset may contain fewer complex patterns or greater noise, limiting its advantage. Naïve Bayes showed an unusual trend—despite its AUC dropping from 0.72 to 0.52, its accuracy rose to 60%, possibly due to better threshold behavior even with weaker class separation. Logistic regression, however, saw declines in both metrics, pointing to growing non-linearity or distribution shifts in the data that hinder linear models. Compared to the previous two tables, the performance changes in this third set were relatively moderate.

This result is surprising in that various types of eating disorders are more prevalent in children with autism, leading to restricted and/or unique eating patterns. Nevertheless, as shown in Figure 3, relying only on exogenous metabolites, many of which originate in food, is less efficient for distinguishing between the ASD and control groups.

### 3.4. Behavioral Classifiers

Having established metabolomic signature patterns, we investigated whether these biochemical findings correspond with behavioral classifications in the same cohort. This analysis is crucial for understanding the potential relationship between urinary metabolite profiles and clinical behavioral presentations in ASD children. In addition to the urine metabolites, we explored the effectiveness of the behavioral classifiers, which were determined from questionnaires that the parents had completed, in distinguishing between the ASD and control groups compared to the metabolite classifiers. This analysis included questions from the SDQ (Strengths and Difficulties Questionnaire) and the ATEC (Autism Treatment Evaluation Checklist). The random forest classifier successfully differentiated children with ASD from controls based on responses to the Autism Treatment Evaluation Checklist (ATEC), achieving an accuracy of 94% and an area under the curve (AUC) score of 0.95. The key features for distinguishing the groups identified through ATEC analysis were drawn from both the health and behavior subscales: the child being “hooked” or fixed on certain objects/topics and the child “shouting or screaming” (Figure 4A,B, respectively). Notably, the first question alone accounted for 22% of the differentiation for ASD. Figure 4A,B illustrate the distribution of symptoms based on parent-reported severity of ATEC symptoms. It is not surprising that ATEC could distinguish between neurotypical children and children with autism with high accuracy. However, we highlight the options of several specific behaviors that are more indicative than others in the child’s behavior. In the future, these classifiers may be able to predict dysregulation in specific biological pathways. Further examination of these topics is warranted, including correlations with comorbid symptoms, specific behaviors, functioning levels, sex, and age, which could better classify subgroups of children with autism and prompt better treatment targets.

Additionally, the random forest classifier distinguished children with ASD from controls based on Strengths and Difficulties Questionnaire (SDQ) data, with an accuracy of 84% and an area under the curve (AUC) of 0.95. The visualization of the J48 pruned decision tree, which is based on SDQ data, is shown in Figure 4.

Children with autism are known to exhibit atypical eating behaviors such as picky eating, pica, etc. [18]. As such, a major hypothesis might have centered around the fact that metabolites that originate from diet, i.e., food metabolites, would be different in children with autism vs. neurotypical children owing to lack of variety in the diet or owing to higher consumption of processed foods or foods high in sugar instead of whole foods. However, as shown in Figure 3A,B, the endogenous metabolites demonstrated superior classification performance for autism diagnosis compared to the exogenous metabolites, as evidenced by both higher AUC and accuracy values, although many of the endogenous metabolites also implied the influence of microbial interactions. One of the main drivers of differences in digestion is the gut microbiome. Children with autism have different fecal metabolite profiles than neurotypical children do [34,35], as well as increased gastrointestinal comorbidities [36,37] and distinct fecal metabolite profiles with fecal transplants as a successful treatment of symptoms [38,39]. Specifically, the link between the gut and brain has been examined in autism [39] because of the interaction between serotonin dysregulation in autism [17,40,41] and extensive serotonin production in the gut [41,42], which was also noted in this study with the 5-HIAA classifier.

Our results emphasize that in addition to the importance of what the child is eating, the way the child’s body utilizes these nutrients, including the microbiome’s composition, should also be highlighted. While this challenge may be obvious in picky eaters, it should also be considered in children with autism who eat well but may have digestive issues affecting macronutrient breakdown, absorption, or nutrient transport to the brain. For example, many beneficial bacteria are necessary to break down polyphenol-rich foods to be available as regulators of gene expression [43,44]. In contrast, invasive bacteria can obstruct the complete breakdown of proteins and, as a result, interfere with the downstream catabolism of necessary proteins and enzymes [45].

These results add to the broader understanding of differences in food digestion, gut motility, and nutrient absorption in the ASD population with gastrointestinal issues, which are prominent comorbidities of autism [46,47]. Equally interesting are metabolites that serve as strong classifiers, which are also related to precursors of glutathione, such as N-acetyl-cysteine, which has already been explored clinically in autism [27,29,48], and glycine, both of which are important for the detoxification of the body. The high classification accuracy of urine metabolites leads to questions about the possibility of using urine for the diagnosis of autism or other neurological conditions.

The findings of this study suggest an intriguing relationship between urinary metabolomic profiles and behavioral manifestations in individuals with ASD. While behavioral assessments remain the cornerstone of ASD diagnosis, their subjective nature and variability across different contexts pose challenges for consistent classification. Figure 5 presents a detailed comparison of the results between classification by behavioral and metabolite classifiers. This comparison highlights the complementary relationship between these two types of classifiers. In several cases, the behavioral classifier fails to predict ASD, but the metabolite classifier succeeds, and vice versa, exploring the possibility of the benefit of joint use of metabolites and behavior.

As shown in Figure 5, our metabolomic analysis potentially provides a complementary biological signature that aligns with behavioral presentations. A previous study [7] characterized a metabolic signature of ASD and evaluated multiple analytical methodologies to develop predictive tools for diagnosis and disease monitoring. However, their dataset was smaller than ours, focusing on targeted analysis, whereas we employed untargeted polar profiling. Interestingly, metabolites such as indoxyl sulfate, N-α-acetyl-l-arginine, and phenylacetylglutamine were included in both studies, but they did not show significant differences in our results. This could be due to the smaller sample size in their study, or potentially due to differences in genetics and diet, as our study and theirs were conducted in different countries.

A key advantage of our approach is that it examines multiple metabolites simultaneously and integrates behavioral classifiers within the same cohort. This opens the possibility to a better understanding of how biochemical alterations may underpin the behavioral phenotypes observed in individuals with ASD and expands the horizons for those with other neurological conditions, such as ADHD or neurodevelopmental disorders with overlapping heterogeneity in symptoms and metabolomics. This would improve diagnostic accuracy and provide insights into the shared and distinct biological pathways underlying these disorders, potentially guiding more targeted and personalized interventions. This dual approach could pave the way for more accurate diagnostic strategies and personalized therapeutic interventions, addressing both the biological and behavioral aspects of the condition.

## Figures and Tables

**Figure 1 metabolites-15-00332-f001:**
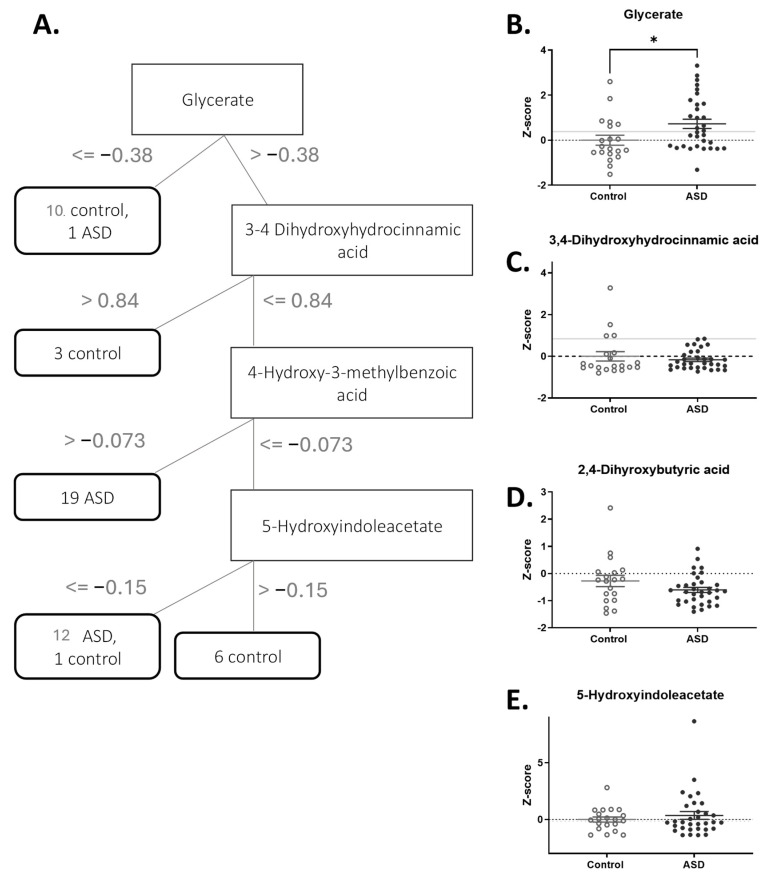
The most significant metabolites for differentiating between ASD patients and controls were identified from a set of 293 metabolites. (**A**). A decision tree generated by the J48 classifier for all metabolites. The nodes in the tree are represented by rectangles and describe the rules for decision-making based on metabolites. The values on the arcs are measured in z-score units. The decision nodes for the ASD and control groups are represented by ellipses. Each decision node contains the number of samples within that node, split between the ASD and control groups. (**B**–**E**). Z-score distributions of the most significant metabolites. The asterisk in 1B indicates a statistically significant difference in glycerate levels between the ASD and control groups.

**Figure 2 metabolites-15-00332-f002:**
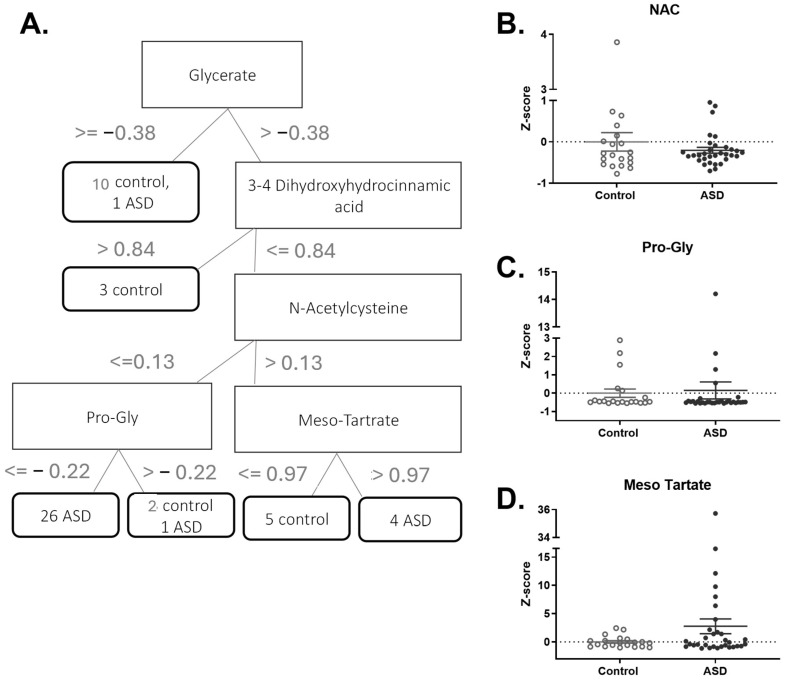
The most significant metabolites for differentiating between ASD patients and controls were identified from a set of 189 endogenous metabolites. (**A**). Decision tree generated by the J48 classifier for 189 endogenous metabolites. The nodes in the tree are represented by rectangles and describe the rules for decision-making based on metabolites. The values on the arcs are measured in z-score units. The decision nodes for the ASD and control groups are represented by ellipses. Each decision node contains the number of children within that node, split between ASD and control. (**B**–**D**) Z-score distributions of the most significant metabolites among the 189 endogenous metabolites.

**Figure 3 metabolites-15-00332-f003:**
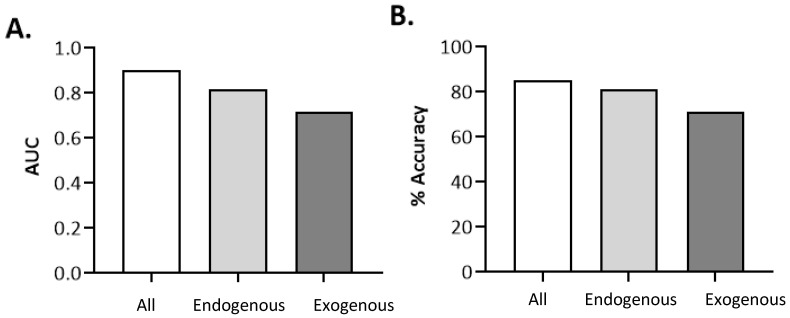
Performance comparison of random forest models trained on different metabolite groups. (**A**) Area Under the ROC Curve (AUC) for models using all metabolites, only endogenous metabolites, and only exogenous metabolites. (**B**) Classification accuracy (%) for the same three metabolite sets. The model trained on all metabolites achieved the highest AUC and accuracy, followed by endogenous, with exogenous metabolites showing the lowest performance in both metrics.

**Figure 4 metabolites-15-00332-f004:**
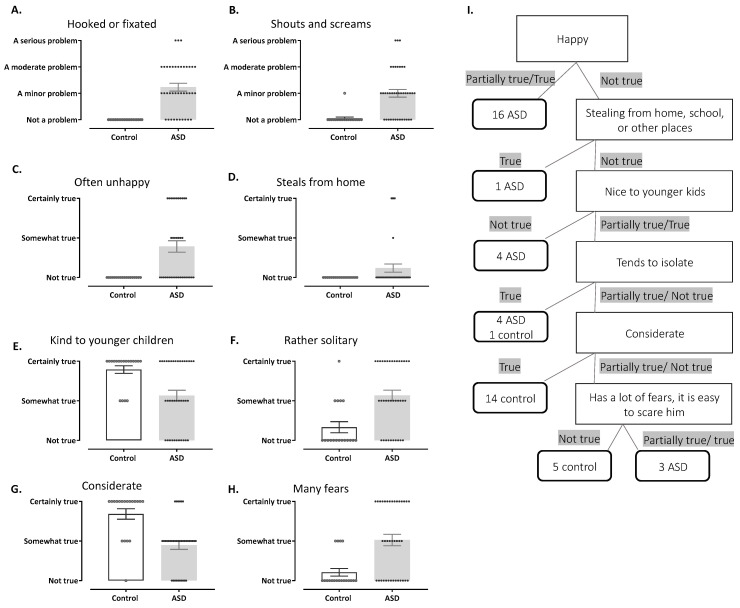
The most significant questions for differentiating between ASD patients and controls based on the ATEC and SDQ questionnaires. (**A**–**H**). The distribution of answers to the most significant questions from the ATEC (**A**,**B**) and SDQ (**C**–**H**) questionnaires. (**I**). A decision tree generated by the J48 classifier for SDQ data. The nodes in the tree are represented by rectangles and describe the rules for decision-making based on the responses. The decision nodes for the ASD and control groups are represented by ellipses. Each decision node contains the number of samples within that node, split between ASD and control groups. “Happy” refers to the level of happiness, where we observe negative behavior in children with ASD. For example, the happiness level was lower in children with ASD. Similarly, negative behavior was observed in children with ASD regarding “Nice to younger kids” and “Considerate”.

**Figure 5 metabolites-15-00332-f005:**
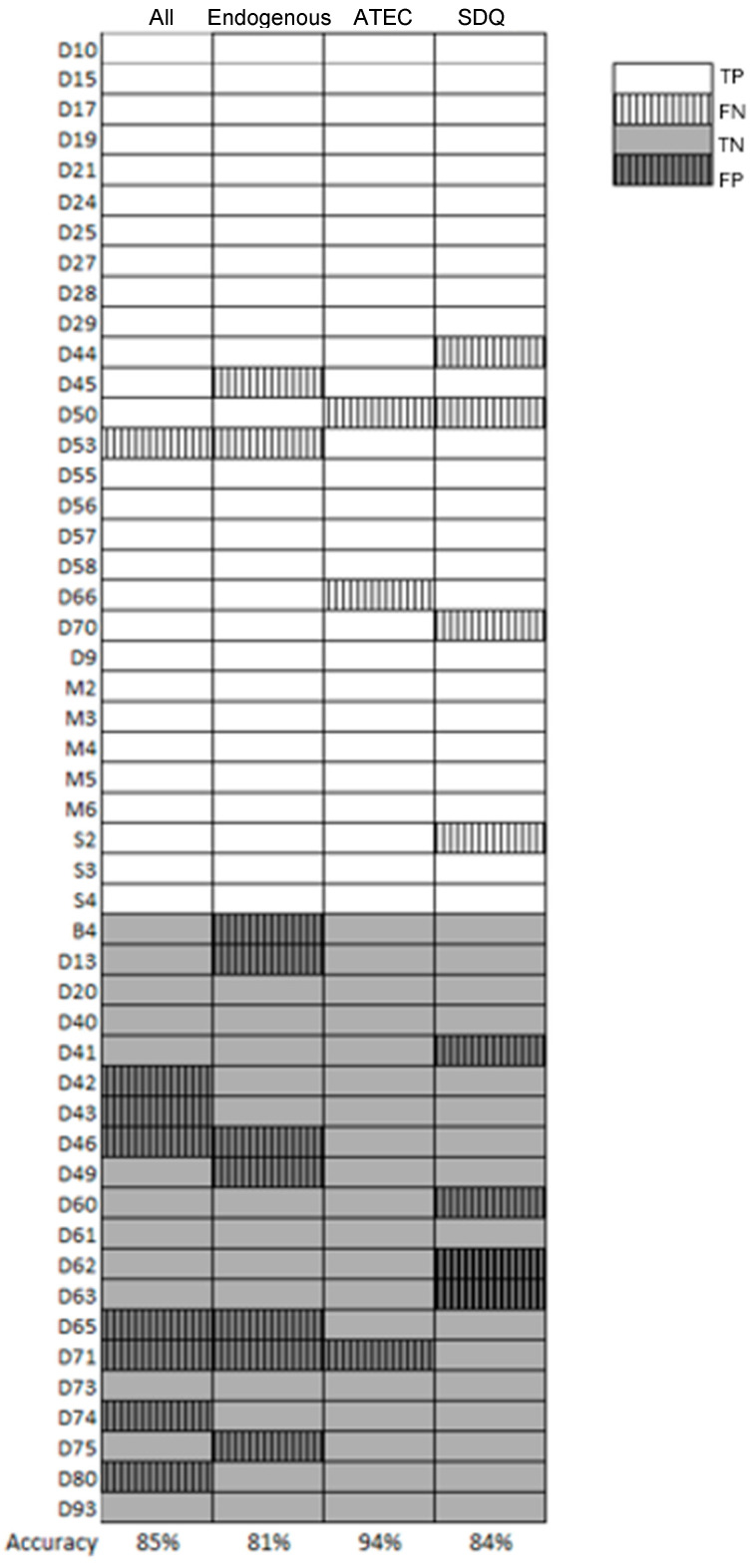
Comparison of classification results for urine metabolites, endogenous metabolites, and behavioral classifiers (ATEC and SDQ). Each row in the heatmap represents a different child in our cohort. The heatmap consists of four columns. The first column, on the far right, displays the classifier’s results for all metabolites. The next column shows the results for endogenous metabolites, followed by columns that describe the classification results based on ATEC and SDQ. In each cell, TP (True Positive) indicates a correct ASD prediction, while TN (True Negative) indicates a correct Control prediction. FN (False Negative) represents instances where an ASD case was misclassified as Control, and FP (False Positive) represents instances where a Control case was misclassified as ASD.

## Data Availability

The data presented in this study are available on request from the corresponding author due to an ongoing additional analysis.

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
