# Peer review of "Urine Metabolomic Profiling and Machine Learning in Autism Spectrum Disorder Diagnosis: Toward Precision Treatment"

_metabolites, 2025, doi:10.3390/metabo15050332_

Round 1

Reviewer 1 Report

Comments and Suggestions for Authors

The manuscript “Urine Metabolic Profiling and Machine Learning...”  is an excellent contribution to the journal, Metabolites.  Three things stand out about this paper:  First, the authors have done an excellent job of handling the data in that the accuracy of the model is not inflated; importantly, their feature selection method was biologically based (internal vs external metabolites), and this process is robust, avoiding inflated accuracies that are common in many metabolomics/machine learning studies.  This reviewer is highly appreciative of their implementation of high-qualiity methods.  Secondly, the finding, that urinary metabolites provides a rather clear discrimination between these two groups, is clinically important, and this could be the foundation for an easier (perhaps earlier?) test for this condition.  Third, I really liked the inclusion of the behavioral data. Congratulations, authors, on an excellent study!

I have just a few suggestions.

  • Can you say something about the hyperparameters for these models? The decision tree-based models were presumably used with the base (unmodified) hyperparameters, but stating that in the experimental would add clarity.
  • In assessing which metabolites are most significant, it’s not clear whether this finding was true across the different models or if you only looked at the J48 classifier. This part could be improved if the authors would specify whether this finding was consistent with all four classifiers and/or if they just looked at overall levels (I think this may be the case.)
  • Something is definitely wrong with Figure 3. The caption does not explain the difference between “A and B”.  In fact, the figure caption would make more sense if there was only 1 bar graph, not two.  And then the text describing Figure 3 (~line 256) says that the graph is for machine learning results based on data from different behavioral questionaires, which also doesn’t mesh with the picture.
  • Another problem with text/figure mismatch is line 287, which refers to specific metabolites correlated with the disease and refers to Fig 3. In this case, I think they may have meant a different fig # all together.
  • Not a complaint, but a compliment: I really like how you included the machine learning results on the behavioral data.  That was a really nice addition.
  • Another compliment: while it’s always ideal to have “lots” of samples and a validation set, I appreciate that these samples are hard to get, and I was impressed with the size of the data set the authors were able to get and the strategies they used to maximize information from them.  Nice work.

Author Response

Comment 1:

  • Can you say something about the hyperparameters for these models? The decision tree-based models were presumably used with the base (unmodified) hyperparameters , but stating that in the experimental would add clarity.

Response 1:

Thank you for your valuable feedback regarding the hyperparameter settings in our experimental section. You're correct that we should be more explicit about the hyperparameter configurations used in our models.

In our revised manuscript, we will add the following clarification to the experimental methods section:

" All classification models were implemented using the WEKA platform 3.8.6, with default (unmodified) hyperparameters applied to all algorithms.  We chose to use these base (unmodified) hyperparameters to provide a standardized evaluation across models without introducing potential advantages through model-specific tuning."

Comment 2:

  • In assessing which metabolites are most significant, it’s not clear whether this finding was true across the different models or if you only looked at the J48 classifier. This part could be improved if the authors would specify whether this finding was consistent with all four classifiers and/or if they just looked at overall levels (I think this may be the case.)

Response 2:

We appreciate the reviewer’s observation. Thank you for highlighting this important point about the feature selection process. We understand the need for clarity regarding how we determined the most significant metabolites across different models. To address this concern, we have added an entirely new paragraph to the Methods section in our revised manuscript that was not present in the original submission. This new paragraph titled "Feature selection" thoroughly explains our approach:

We performed feature selection using all four classifiers (Random Forest, J48 pruned tree, Naive Bayes, and Logistic Regression). Initially, the J48 pruned tree identified four key metabolites, which prompted us to also extract the top four features from each of the other three classifiers. We then conducted a comprehensive cross-evaluation by using each set of four metabolites (from each classifier) as input features for all four classification algorithms. Our analysis showed that the four metabolites selected by the J48 pruned tree (Glycerate, 3,4-Dihydroxyhydrocinnamic acid, 3,4-Dihydroxybenzoate, and 5-Hydroxyindoleacetate) yielded the best classification performance consistently across all classifiers.We also evaluated the combined set of all 16 unique top-ranked features (4 from each classifier), but this broader selection did not improve classification performance compared to using just the four J48-selected metabolites. The finding that the J48-selected metabolites performed best was consistent across all four classification models, not just the J48 classifier itself. This cross-validation approach strengthens our confidence in these four metabolites as truly significant biomarkers.

We believe this additional information in our methods section addresses the reviewer's concern by clarifying that we systematically evaluated feature importance across all classifiers rather than relying solely on one model's assessment.

Comment 3:

  • Something is definitely wrong with Figure 3. The caption does not explain the difference between “A and B”.  In fact, the figure caption would make more sense if there was only 1 bar graph, not two.  And then the text describing Figure 3 (~line 256) says that the graph is for machine learning results based on data from different behavioral questionaires, which also doesn’t mesh with the picture.

Response 3:

Thank you for pointing out the issues with Figure 3 and its description. You are absolutely right that there was a lack of clarity in both the caption and the corresponding text in the manuscript.

We have completely revised the figure caption to properly explain the difference between panels A and B:

"Figure 3. Performance comparison of Random Forest models trained on different metabolite groups. (A) Area Under the ROC Curve (AUC) for models using all metabolites, only endogenous metabolites, and only exogenous metabolites. (B) Classification accuracy (%) for the same three metabolite sets. The model trained on all metabolites achieved the highest AUC and accuracy, followed by endogenous, with exogenous metabolites showing the lowest performance in both metrics."

Additionally, we have corrected the text in line 256 by changing the reference from (Fig 3A, B) to (Figures 4A, 4B respectively) to ensure consistency with the figure. The revised text now properly describes the comparison of model performance across different metabolite groups (all, endogenous, and exogenous) as shown in Figure 3.

Thank you for catching this discrepancy. The corrected figure caption and text should now present a coherent explanation of the Random Forest model performance across the different metabolite groupings.

Comment 4:

  • Another problem with text/figure mismatch is line 287, which refers to specific metabolites correlated with the disease and refers to Fig 3. In this case, I think they may have meant a different fig # all together.

Response 4:

Thank you for highlighting the potential mismatch between the text on line 287 and Figure 3. I understand your concern about the reference to Figure 3 when discussing metabolites correlated with autism.

To clarify, we do indeed intend to reference Figure 3 in this context. Figure 3 shows the performance comparison of Random Forest models trained on different metabolite groups. The text on line 287 correctly interprets the results displayed in Figure 3, noting that "the endogenous metabolites, were better classifiers for autism diagnosis than the exogenous metabolites were." This conclusion directly relates to the performance metrics shown in Figure 3, where the model using only endogenous metabolites performs better than the model using only exogenous metabolites.

To improve clarity and avoid confusion, we will revise the text to more explicitly connect to the specific panels in Figure 3:

"However, as shown in Figure 3A and 3B, the endogenous metabolites demonstrated superior classification performance for autism diagnosis compared to the exogenous metabolites, as evidenced by both higher AUC and accuracy values, although many of the internal metabolites also implied the influence of microbial interactions."

Thank you for your careful review, which has helped us ensure consistency between our text and figures.

Reviewer 2 Report

Comments and Suggestions for Authors

The study investigates potential diagnostic markers for Autism Spectrum Disorder (ASD) in children's urine using various machine learning techniques. A total of 293 metabolites were identified and categorized into 188 internal (endogenously produced) and 105 external (exposome-originated) metabolites. The work demonstrates the potential utility of urinary metabolomic profiling as a complementary diagnostic tool for ASD. However, several issues require clarification and revision prior to publication. Specific comments follow:

  1. The abbreviation "ASD" in the title is inappropriate. It should be written as "Autism Spectrum Disorder".
  2. Please clarifywhy negative ionization mode was exclusively selected for profiling the polar phase of urine.

3.The authors should clarify why only polar metabolites were analyzed by LC-MS. A brief discussion about the limitations of this approach for comprehensive metabolomic coverage would be valuable.

4.The LC-MS conditions are incompletely described, particularly regarding the MS instrumentation parameters. Please provide full details of the data acquisition mode and other critical MS settings.

5.In the statement: "Relative levels of 12 detected metabolites were normalized to the internal standards and creatinine": Clarify the methodology for creatinine detection in participant samples. Specify the internal standards used, this information should be added to the Methods section.

6.Please clarify the relationship between conventional metabolomic data processing methods (PCA, PLS-DA, OPLS-DA) and the machine learning techniques employed in this study. A comparative discussion of their respective advantages would strengthen the methodology section.

7.The terminology "internal metabolites and external metabolites" is non-standard. It is recommended to use the more widely accepted terms "endogenous metabolites" and "exogenous metabolites" respectively.

  1. The comparative performance of different machine learning techniques requires clearer presentation.
  2. The manuscript lacks essential method validation data. Please include:Accuracy and precision assessments for both sample preparation protocols and detection methods. Quality control measures implemented during analytical procedures.

Author Response

Comment 1:

  1. The abbreviation "ASD" in the title is inappropriate. It should be written as "Autism Spectrum Disorder".

Response 1:

Thank you for the comment. We agree with the reviewer’s suggestion and have updated the title to spell out "Autism Spectrum Disorder" in full, instead of using the abbreviation "ASD," to ensure clarity and formality.

Comment 2:

  1. Please clarify why negative ionization mode was exclusively selected for profiling the polar phase of urine.

Response 2:

We appreciate the reviewer raising this important methodological question. The in-house metabolite library was prepared and optimized in negative ionization mode. Therefore, this ionization mode was used for urine polar metabolite profiling.

Comment 3:

3.The authors should clarify why only polar metabolites were analyzed by LC-MS. A brief discussion about the limitations of this approach for comprehensive metabolomic coverage would be valuable.

Response 3:

We were interested in polar metabolites since most of the existing literature is based on polar metabolites, and as a first step, we wanted to see where our cohort is compared to the available data. Many papers used NMR for polar metabolites. However, we chose LC-MS as a non-targeted approach to obtain a larger number of metabolites. In addition, polar metabolites provide data regarding many biological processes that are known or suspected to be altered in some of the children with developmental disorders.

We are aware that there is a possibility of also identifying semipolar metabolites and lipid metabolites. The reviewer is correct that these would be very interesting directions for the future. However, this study requires different running settings and added expenses.

Comment 4:

4.The LC-MS conditions are incompletely described, particularly regarding the MS instrumentation parameters. Please provide full details of the data acquisition mode and other critical MS settings.

Response 4:

We appreciate the reviewer’s comment regarding the MS parameters. Full details of the data acquisition mode and critical MS settings have now been provided in the Methods section of the revised manuscript.

Comment 5:

5a In the statement: "Relative levels of 12 detected metabolites were normalized to the internal standards and creatinine": Clarify the methodology for creatinine detection in participant samples. 5b Specify the internal standards used, this information should be added to the Methods section.

Response 5:

5a We thank the reviewer for highlighting the need to clarify the creatinine normalization methodology. This information has been added to the Methods section of the revised manuscript.

5b This information was specified in the method in the article” containing the following internal standards: C13- and N15-labeled amino acid standard mixtures (Sigma, 767964)“.

Comment 6:

6.Please clarify the relationship between conventional metabolomic data processing methods (PCA, PLS-DA, OPLS-DA) and the machine learning techniques employed in this study. A comparative discussion of their respective advantages would strengthen the methodology section.

Response 6:

Thank you for the insightful comment regarding conventional metabolomic data processing methods and their relationship to machine learning techniques in our study.

Principal Component Analysis (PCA), Partial Least Squares-Discriminant Analysis (PLS-DA), and Orthogonal PLS-DA (OPLS-DA) are widely used in metabolomic studies to reduce dimensionality, visualize data structure, and enhance group separability. These methods primarily focus on identifying underlying patterns in complex datasets, offering insights into metabolite distributions and biological variability.

The advantage of conventional methods like PCA and PLS-DA lies in their ability to reveal key discriminative features and help reduce noise prior to classification.

In our study, PCA and clustering algorithms were initially employed to explore metabolite variance and detect potential outliers. However, supervised machine learning models were ultimately chosen for classification due to their superior ability to handle complex, high-dimensional data while maximizing accuracy and AUC values.

We incorporated this comparative discussion into the methodology section to strengthen the manuscript and enhance clarity regarding the analytical framework.

Comment 7:

7.The terminology "internal metabolites and external metabolites" is non-standard. It is recommended to use the more widely accepted terms "endogenous metabolites" and "exogenous metabolites" respectively.

Response 7:

Thank you for the helpful suggestion. We agree that "endogenous metabolites" and "exogenous metabolites" are more appropriate and widely accepted terms. We have updated the terminology throughout the manuscript accordingly, replacing "internal metabolites" with "endogenous metabolites" and "external metabolites" with "exogenous metabolites."

 Comment 8:

  1. The comparative performance of different machine learning techniques requires clearer presentation.

Response 8:

Thank you for your comment regarding the need for clearer presentation of the comparative performance of different machine learning techniques. I would like to clarify that Tables S1, S2, and S3, which were included in the original submission, already contain all the detailed results of comparisons between the different classifiers. These tables provide comprehensive metrics for each classifier's performance.

To address your comment regarding the need for clearer presentation of the comparative performance of different machine learning techniques, we have added comprehensive explanations in the results section that detail the performance differences between classifiers and provide context for these variations.

These additions provide a clearer presentation of the comparative performance, including specific metrics, analysis of why certain models performed better or worse with different metabolite subsets, and interpretations of these performance differences.

Comment 9:

  1. The manuscript lacks essential method validation data. Please include Accuracy and precision assessments for both sample preparation protocols and detection methods. Quality control measures implemented during analytical procedures.

Response 9:

We appreciate the reviewer’s comment regarding method validation. Several quality control measures were implemented to ensure the accuracy and precision of both sample preparation and detection procedures, supporting the robustness and reproducibility of the analytical workflow. These details have been described now in the revised Methods section.

Round 2

Reviewer 2 Report

Comments and Suggestions for Authors

There is no further suggestion.